# Genomic Analysis of 18th-Century Kazakh Individuals and Their Oral Microbiome

**DOI:** 10.3390/biology10121324

**Published:** 2021-12-14

**Authors:** Anna E. White, Toni de-Dios, Pablo Carrión, Gian Luca Bonora, Laia Llovera, Elisabetta Cilli, Esther Lizano, Maral K. Khabdulina, Daniyar T. Tleugabulov, Iñigo Olalde, Tomàs Marquès-Bonet, François Balloux, Davide Pettener, Lucy van Dorp, Donata Luiselli, Carles Lalueza-Fox

**Affiliations:** 1Institute of Evolutionary Biology, CSIC-Universitat Pompeu Fabra, 08003 Barcelona, Spain; annaewhite6@gmail.com (A.E.W.); antonio.dedios01@estudiant.upf.edu (T.d.-D.); pc.quilis@gmail.com (P.C.); laia.llovera@upf.edu (L.L.); esther.lizano@upf.edu (E.L.); inigo.olalde@gmail.com (I.O.); tomas.marques@upf.edu (T.M.-B.); 2Estonian Biocentre, Institute of Genomics, University of Tartu, 51010 Tartu, Estonia; 3ISMEO—International Association for Mediterranean and East Studies, 00186 Rome, Italy; eclglbonora@gmail.com; 4Department of Cultural Heritage, University of Bologna, 48121 Ravenna, Italy; elisabetta.cilli@unibo.it; 5Institut Català de Paleontologia Miquel Crusafont, Universitat Autònoma de Barcelona, 08193 Cerdanyola del Vallès, Spain; 6K.A. Akishev Institute of Archaeology, L.N. Gumilev Eurasian National University, Nur-Sultan 010000, Kazakhstan; mk_khabdulina@mail.ru (M.K.K.); d.tleugabulov@gmail.com (D.T.T.); 7Centro de Investigación “Lascaray” Ikergunea, BIOMICs Research Group, Universidad del País Vasco, 01006 Vitoria-Gasteiz, Spain; 8Catalan Institution of Research and Advanced Studies (ICREA), 08010 Barcelona, Spain; 9CNAG-CRG, Centre for Genomic Regulation (CRG), Barcelona Institute of Science and Technology (BIST), 08036 Barcelona, Spain; 10UCL Genetics Institute, Department of Genetics Evolution & Environment, University College London, London WC1E 6BT, UK; f.balloux@ucl.ac.uk; 11Department of Biological, Geological and Environmental Sciences, University of Bologna, 40126 Bologna, Italy; davide.pettener@unibo.it

**Keywords:** paleogenomics, ancient pathogens, Central Asian steppe, red complex, bacteria

## Abstract

**Simple Summary:**

Paleogenomic research has enhanced our understanding of past human populations and migrations. Moreover, ancient DNA can also provide information on the interactions between humans and their associated pathogens. Periodontitis, currently suffered by millions of people worldwide, is a gum infection that is caused by pathogenic oral bacteria, widely referred to as the “red complex”, which can lead to tooth loss and has been linked to an increased risk of oesophageal cancer. In this study we combine human population genetics with genomic analysis of the oral microbiome of 18th-century people from the Kazakh Khanate in Kazakhstan to gain insight into their health status and explore bacterial coadaptation to recent dietary changes.

**Abstract:**

The Asian Central Steppe, consisting of current-day Kazakhstan and Russia, has acted as a highway for major migrations throughout history. Therefore, describing the genetic composition of past populations in Central Asia holds value to understanding human mobility in this pivotal region. In this study, we analyse paleogenomic data generated from five humans from Kuygenzhar, Kazakhstan. These individuals date to the early to mid-18th century, shortly after the Kazakh Khanate was founded, a union of nomadic tribes of Mongol Golden Horde and Turkic origins. Genomic analysis identifies that these individuals are admixed with varying proportions of East Asian ancestry, indicating a recent admixture event from East Asia. The high amounts of DNA from the anaerobic Gram-negative bacteria *Tannerella forsythia*, a periodontal pathogen, recovered from their teeth suggest they may have suffered from periodontitis disease. Genomic analysis of this bacterium identified recently evolved virulence and glycosylation genes including the presence of antibiotic resistance genes predating the antibiotic era. This study provides an integrated analysis of individuals with a diet mostly based on meat (mainly horse and lamb), milk, and dairy products and their oral microbiome.

## 1. Introduction

Central Asia is a crucial region to study Eurasian human history due to its pivotal role in east-west population movements. Due to advancements in ancient DNA (aDNA) techniques and thermal conditions that are favourable for DNA preservation, this region is being extensively surveyed. Recent aDNA research has provided an understanding of the main movements of people in Central Asia during the Bronze and Iron Age [1,2,3,4,5]. Large-scale genetic turnovers have been identified across the Central Steppe of Asia, which consists of mainly current-day Kazakhstan, Mongolia, and Russia [2,4,5,6,7]. About 5000 years ago, there was a major migration from the Yamnaya population in the Pontic-Caspian steppe to both Eastern Europe and South Asia [2]. During the Iron Age, the Scythian expansion took place along with migrations from nomad warrior cultures from the Xiongnu empire from the east and the Persian kingdoms from the south. This was followed by additional population movements after the collapse of the Mongol empire [5,8,9,10]. However, recent population movements, spanning the last centuries, have not been systematically addressed in these studies.

Kazakhstan is the largest Central Asian country and constitutes a land bridge between Europe and Asia. Due to its central location, the genetic structure of its peoples is of significance to understand the complexities of the steppe’s past migrations [11]. In 2018, the excavation of an early Iron Age burial mound of Kuygenzhar, a royal cemetery located about 19 km south-east from Nur-Sultan (former Astana) on the right bank of the river Ishim, uncovered the remains of several individuals of uncertain cultural attribution (Appendix A). The original top of the kurgan was disturbed to accommodate the graves of seven people that were buried adhering to the Islamic ritual in depths between 0.85 and 1.15 metres. The graves were oriented in a north-west to south-east line; the bodies were lying on their right side and had neither evidence of funerary goods nor coffin. The skeletal remains, corresponding to adult people, had been radiocarbon-dated to 1713–1785 CE (Appendix A), which places these individuals at the times of the Kazakh Khanate (from 1465 to 1847). This khanate was founded in the mid-fifteenth to sixteenth century as a union of nomadic tribes of Turkic and Mongol origins [12].

To understand the genetic ancestry of these individuals in their historical context, we generated genome-wide data from teeth samples. Together with analysis of the human genetic data, we also screened for DNA that was associated with the oral microbiome. The composition of bacteria that was present in the oral microbiome can lead to chronic dental diseases such as periodontitis that can cause tooth loss and severe damage to the alveolar bone [13,14]. Oral bacteria that cause periodontitis are commonly referred to as the “red complex”, which includes *Treponema denticola, Porphyromonas gingivalis*, and *Tannerella forsythia.* Bacteria that are associated with the “red complex” have also been linked to a higher risk of esophageal cancer [15,16], commonly associated with diabetes mellitus [17,18], and suggested as a risk factor for multiple other syndromes [19,20,21], illustrating the importance of studying these species. Interestingly, according to historical accounts, the diet of the people during the Kazakh Khanate was the traditional steppe diet that was based on meat and dairy products, prior to the introduction of a carbohydrate-based diet.

Paleogenomic approaches have the potential to shed new light on the evolutionary processes that influenced the spread of pathogens and diseases in the past [22], which is of particular relevance at hotspots of population turnover and encompassing time periods predating major population health and dietary shifts.

## 2. Materials and Methods

### 2.1. Ancient DNA Extraction

DNA was extracted and sequenced from dental material that was sampled from five Kuygenzhar individuals in the Paleogenomics laboratory at the Institute of Evolutionary Biology in Barcelona. A cementum powder (50 mg) sample was obtained from the teeth’s roots with a Dremel drilling machine and subjected to DNA extraction. DNA extraction was conducted following the method that is described in Dabney et al. [23]. A total of 1 ml of digestion buffer (final concentrations: 0.45 M EDTA, 0.25 mg/mL proteinase K) was added to each sample -including an extraction blank- and was incubated overnight at 37 °C on a rotating device (750–900 rpm). The undigested powder was then pelleted by centrifugation in a bench-top centrifuge for 2 min at maximum speed (16,100× *g*); the supernatant was added to 10 mL of binding buffer (final concentrations: 5 M guanidine hydrochloride, 40% (v/v) isopropanol, 0.05% Tween-20, and 100 mM sodium acetate (pH 5.2)) and purified on a High Pure Extender column (Roche). The resulting DNA extracts were finally eluted with 45 μL of low EDTA TE buffer (pH 8.0) and quantified using a Qubit instrument.

Following extraction, 10–20 ng of DNA extract was converted into Illumina sequencing libraries following the BEST protocol [24]. Each library was amplified by PCR using two uniquely barcoded primers. After index PCR, the libraries were purified with a 1.5x AMPure clean (Beckman Coulter, Brea, CA 92821, USA) and eluted in 25 μL of low EDTA TE buffer (pH 8.0). Libraries were quantified using BioAnalyzer and sequenced by HiSeq 4000 (Illumina, San Diego, CA 92122, USA).

### 2.2. Processing of Sequencing Data and Mapping to the Human Genome

Adapter sequences were trimmed from the reads with AdapterRemoval v.2.2.2 [25] prior to mapping. The reads were further filtered for a minimum quality of 3 and a minimum length of 30 bp. For whole genome analysis, the sequencing reads of at least 30 bp were mapped against the human reference genome GRCh37 with the Burrows-Wheeler Aligner (BWA) v.0.1.17 backtrack algorithm [26] with an edit distance parameter (-n) of 0.01 and a gap open parameter (-o) of 2. The mapped reads were merged and sorted after which the duplicates were removed using picard v.2.25.2 [27]. The Mapped reads were filtered for a mapping quality of at least 30 with SAMtools v.1.12[28]. Basic mapping statistics were determined with Qualimap [29]. The coverage was estimated using 1 kb windows using bedtools v.2.24.0 [30] and visualised using Circos v.0.69-8 [31].

### 2.3. Assessing the Authenticity of the Human Genomic Data

Since DNA degrades over time, authentic aDNA reads show signs of characteristic decay that can be quantified. Beyond DNA fragmentation, post-mortem damage at the ends of the DNA molecules can result in cytosine (C) to thymine (T) and guanine (G) to adenine (A) nucleotide changes in the 5′ end and 3′ end, respectively [32,33], becoming more frequent towards the end of the DNA fragments [34,35]. For the identification of post-mortem damage PMDtools v.0.60 and MapDamage v.2.0.8 [36,37] were applied.

### 2.4. Assessing the Genetic Sex and Degree of Contamination

The genetic sex of the individuals was established by using the ratio of sequences mapped to the X and Y chromosomes [38]. DNA contamination was estimated for the mitochondrial DNA (mtDNA) and the X chromosome with Schmutzi v.1.5.4 [39] and angsd v.0925 [40], respectively.

### 2.5. Mitochondrial DNA and Y-Chromosome Haplogroup Determination

mtDNA haplotype calling was performed with GATK UnifiedGenotyper v.3.7 [41]. Mitochondrial and Y-Chromosome haplogroups were assigned by manually annotating the variants that were present using HaploGrep2 v.2.4.0 [42] based on Phylotree Build 17 [43] and the International Society of Genetic Genealogy (http://www.isogg.org, accessed on 11 October 2021) version 15.73, respectively.

### 2.6. Kinship Analysis

We determined kinship relationships among the Kuygenzhar samples using the Relationship Estimation from Ancient DNA (READ) program [44]. We assessed the proportion of non-matching alleles (P0) for each pair of individuals and normalized these values by the P0 median value (using the default window-size value of 1 Mbps).

### 2.7. Analysing Population Structure via Clustering Algorithms

aDNA data that were generated from the individuals that were analysed were merged with the Single Nucleotide Polymorphism (SNP) Human Origins (HO) dataset (available at https://reich.hms.harvard.edu/datasets, accessed on 6 July 2021), which includes both ancient and modern DNA samples. Pseudohaploid genotype calls were performed using PileupCaller’s RandomCalling option, available in sequenceTools v.1.2.1 [45]. Several analyses were subsequently performed to understand the genetic ancestry of the Kuygenzhar individuals and how they relate to ancient and present-day Central Asians.

A principal component analysis (PCA) was conducted to place the Kuygenzhar individuals in the context of present-day genetic variation. The HO dataset was first filtered by removing phenotypic SNPs which are under selective pressure. A PCA was performed using the smartpca package of EIGENSOFT v.7.2.1 [46,47] with lsqproject = YES and shrinkmode = YES. In this case, the principal components (PC) were computed using a set of present-day individuals that were genotyped using the Human Origins array [3,48,49]. The Kuygenzhar individuals were then projected onto these components along with other published ancient DNA individuals.

ADMIXTURE v.1.3 program [50] was used to perform a model-based clustering analysis. The same dataset as explained for the PCA was used but SNPs in linkage disequilibrium were discarded from this dataset by means of PLINK program version 1.9 (-indep-pairwise 200 25 0.4) [51]. ADMIXTURE was run computing 10 replicates of clusters from K = 2–20 (with a random seed) and assessing cross-validation error (-cv). High log-likelihood replicates with an average pairwise similarity of over 0.99 were selected and plotted using Pong (version 1.4.1) [52].

### 2.8. Ancestry Modelling

Modelling the ancestry of the Kuygenzhar Kazakhs was carried out using the *f*-statistics framework tests [53]. *f*_4_ statistics were generated using the ADMIXTOOLS v.6.0 built-in module qpAdm [54] to infer possible ancestry mixtures based on patterns of shared drift [55]. The dataset that was used for the mentioned tests consisted on ancient and modern individuals that were genotyped to the ‘1240K’ array (which contained over a total of 1,233,013 SNPs (dataset v44.3.1240K, available at https://reichdata.hms.harvard.edu/pub/datasets/amh_repo/curated_releases/V44/V44.3/SHARE/public.dir/v44.3_1240K_public.tar, accessed on 6 July 2021).

To determine the ancestry components exhibiting best fit to the data that were obtained from the Kuygenzhar individuals we applied the ADMIXTOOLS module qpAdm, using a set of outgroups containing proximal and distal sources of ancestry; with parameter allsnps = YES. The outgroup populations that were used for the analyses were: Mbuti.DG (Central African hunter-gatherers), ONG.SG (indigenous Andamanese islanders), Ami.DG (Taiwanese Aborigines), Mixe.DG (Native Americans), Israel_Natufian_published (Early Holocene Levantine hunter-gatherers Natufian), Iran_GanjDareh_N (Early Neolithic Iranians), Turkey_N (Early Neolithic farmers from western Anatolia), and Italy_North_Villabruna_HG (Pleistocene European hunter-gatherer from Italy) as in [5,56]. Potential source populations were explored using a different set of populations that were close in geographical location and time to the Kuygenzhar individuals in the 1240K dataset [7,8,11,57,58,59]. We explored the viability of several one-way and two-way ancestry-models for each of the samples individually and all the samples grouped as a single population.

### 2.9. Processing of Sequencing Data and Mapping of Ancient Microbes

Microbial DNA that was present in the cementum DNA extracts was assigned using Kraken2 [60,61] against a database of bacteria, archaea, fungi, protozoa, and viruses, after optical duplicates, low complexity reads, and human DNA were removed. Analysis with Kraken2 of all Kuygenzhar individuals revealed a significant presence of DNA reads deriving from bacteria typically considered as “red complex”, especially those assigned to *Tannerella forsythia* [15].

Due to a high presence of *T. forsythia* in our samples we further analysed this bacteria by mapping all non-human reads of at least 30 bp against the reference genome of *T. forsythia* (NC_016610.1) using BWA aln and samse, disabling seeding, setting an edit distance of 0.01, and a gap open penalty of 2 [26]. The mapping procedure was done following the same tools as for the human DNA reads. Estimates of the expected coverage of the data to the reference given read length and read abundance was performed as described in Rasmussen et al. [62]. The mapped *T. forsythia* sequences was visualised with Circos 0.69–9 [31].

For *T. forsythia*, the variants were called with GATK UnifiedGenotyper [41] and filtered using vcftools v.0.1.14 [63], removing indels, positions with a genotyping quality less than 30, and setting a minimum depth of coverage of 5 folds (for B3 and B7) or 2 folds of coverage (for B5 and B6) [63]. In the case of heterozygous positions, the majority allele was considered as homozygous if its allele depth proportion was of 70% or higher; the positions that still remained heterozygous were removed.

For *T. denticola* and *P. gingivalis,* modern and ancient samples were mapped as described for *T. forsythia.* Variants were called using GATK UnifiedGenotyper and filtered using vcftools v.0.1.14. The minimum depth was set to 2 folds of coverage.

### 2.10. Assessing the Authenticity of the Ancient Microbial DNA

As with the human data, deamination at the ends of the DNA molecules was estimated as a signal for the authenticity of aDNA using PMDtools and MapDamage [36,37]. The edit distance—the difference between the reference genome and the Kuygenzhar microbial strains across the mapped genome- was determined with SAMtools [28] to establish whether the reads had been mapped against the correct reference strain. Visualisation of the aDNA damage pattern was performed with R (version 4.0.4) [64].

### 2.11. Phylogenetic Analysis of T. forsythia

With the intention of creating a robust *T. forsythia* phylogeny, we constructed a high-quality SNP reference panel. This panel was created using samples with more than 40% of the reference covered and ≥1x average depth of coverage. This included two of the Ancient Kazakh samples (B3 and B7) and then published data from two ancient Mexican samples, four ancient European samples, one ancient African, thirteen present-day strains, and a strain that was recovered from dog dental plaque (Appendix A). Positions for those samples were called as described for the Kazakh samples. This approach resulted in a dataset comprising of 129,641 SNPs, covered in at least 60% of the samples. Afterwards, consensus *fasta* files of the aforementioned samples were created using the reference *T. forsythia* genome 92A2 and vcftools. Positions that did not pass the filtering criteria (missing, depth, heterozygosity, and genotyping quality), or variant positions outside this dataset were masked using bedtools maskfasta v.2.2.2 [30]. We used this high quality dataset to create a maximum likelihood tree using RAxML v. 8.2 [65]. We set the algorithm GTRGAMMA, with 1000 bootstraps and the OH2617_COT-023 strain that was recovered from a dog as an outgroup based in its basal position in respect to the other strains in the NCBI blast-generated dendrogram (https://www.ncbi.nlm.nih.gov/genome/11045, accessed on 5 November 2021).

In parallel, we created a low-quality dataset. This dataset was generated by calling the 129,641 variant positions in 25 additional samples, filtering them for a minimum genotyping quality of 30, minimum depth of 2, and heterozygous positions with an allele depth proportion of 70%. We merged both low-quality and high-quality datasets and ran RAxML again with the GTRGAMMA algorithm and 100 bootstraps. The results were displayed with ggtree [66] and iTOL [67].

### 2.12. Phylogenetic Dataset Filtering

A requirement for the calculation of a precise tip calibrated dating is the removal of recombinant positions and homoplasies that could substantially alter the branch length of the input phylogeny [68]. To do so, we filtered out positions in recombinant tracks by using ClonalFrameML [69] using our high quality dataset, the subsequent 1000 bootstraps tree, and an estimated kappa value of 4.645. This resulted in the discovery of 6003 recombination tracks across the 23 strains, accounting for 2,410,577 positions, roughly 2/3 of the total reference genome (Appendix A). After purging the dataset of all detected recombinant tracts, we were left with 2252 variable positions.

Using the filtered alignment, we generated a new tree with RAxML (GTRGAMMA; 100 bootstraps; OH2617_COT-023 outgroup). To remove any further undetected inconsistent sites the alignment and resulting tree was input to HomoplasyFinder [70], which assigns a consistency index value to every variant position in the alignment [71]. A total of 758 positions were identified as homoplasic and were subsequently purged resulting in a final dataset of 1494 positions. A further phylogenetic tree was generated based on this input alignment using RAxML (GTRGAMMA; 100 bootstraps; OH2617_COT-023 outgroup).

### 2.13. Tip Calibrated Dating

To infer the age and split times that were associated to the *Tannerella forsythia* clade, BactDating [72] was applied. We first ran a randomised significance test to certify the presence of a temporal signal in the phylogeny. After recovering a statistically significant signal we applied a root–tip analysis, for which samples without a collection date were treated as missing. To find the most suitable model, we ran five different iterations of every model and selected the best model-fit based on their DIC and ESS values. This resulted in selection of the BactDating arc model with 1·10^7^ iterations, yielding an ESS value of 501 and a DIC of 24.60. We also retrieved the convergence statistics (Appendix A). The resulting tip calibrated phylogeny was visualised using the R package ggtree.

### 2.14. Phylogenetic Analysis of T. denticola and P. gingivalis

While the focus of our work was on *T. forsythia*, we additionally considered the other oral pathogens for which sufficient aDNA was recovered; *T. denticola* and *P. gingivalis*. To place these ancient bacterial genomes in a modern context, we selected published samples from both species with more than 1X of average depth [73]. SNP datasets for each bacterium were created considering only variant positions covered in more than 75% of the samples. The phylogenetic trees were created using RAxML model GTRCAT with 100 bootstraps. The results were visualised using iTOL [67] (Appendix A).

### 2.15. T. forsythia Pan-Genome Analysis

To explore the evolutionary history of the acquisition of specific genes in *T. forsythia*, the differential presence of all annotated genes was assessed in four ancient European and twelve ancient Mexican individuals [15,74,75], along with modern strains. A total of 36 virulence, 19 glycosylation, and 24 antibiotic resistance genes from a comprehensive custom panel that were previously published [76] were mapped against all samples in the dataset for which the coverage was determined with BEDtools [30]. Heatmaps were created using R (version 4.0.4) [64].

## 3. Results

### 3.1. Human DNA Analysis

#### 3.1.1. Assessing aDNA Reliability

The authenticity of ancient human DNA was determined by the presence of postmortem damage patterns at the ends of DNA reads (Appendix A). Of the five samples, four samples of the Kuygenzhar individuals yielded enough data for further population genetics analyses (Table 1, coverage > 0.1x). Of these four, three were genetically male and one—individual B2_B3- was determined as genetically female (Appendix A). The X chromosome heterozygosity showed low estimates of modern human contamination for the three male samples (Appendix A). At the mtDNA only B2_B6 showed significant amounts of contamination in one genomic library that was subsequently removed from the analyses (Appendix A).

#### 3.1.2. Uniparental Markers and Kinship Relationships

The mtDNA haplogroups differed in all four individuals, although all of the observed haplogroups can be considered common in sampled current-day Kazakh populations [77] (Appendix A). These haplogroups suggest that the local maternal ancestry make-up of these individuals has persisted into current-day Iran and Eastern Asia [77]. For the three males with sufficient data, two different Y-chromosome haplogroups were determined (Appendix A). Individual B2_B5 had a G1b-L830 haplotype, which is the most common in modern-day Kazakhs -mainly in the Argyn tribe- and likely originated in the region of current-day Iran, where the highest diversity is found [78]. Their most recent common genealogical ancestor is estimated to have lived 600 years ago [78]. B2_B6 and B2_B7 conversely were assigned to the C2 haplogroup, with the former having enough coverage for the assignation to C2a1a3-M504 lineage. This haplotype is frequent in present Mongols and east Kazakhs and is one of the founding paternal lineages of Mongolic-speaking populations [79,80]. Hence, uniparental markers suggest a certain amount of diversity in the ancestry of the Kuygenzhar individuals.

Kinship analysis identified a first-degree relationship between individuals B2_B6 and B2_B7 (Appendix A). Both are considered genetically male with mitochondrial haplogroups U5a1a1h and T2b2, respectively; therefore, it can be determined this is a father and son relationship. However, we cannot know who is the father and who is the son, as they are both adult males around 25–35 years-old.

#### 3.1.3. Human Genetic Data Exploration

We applied principal component analysis (PCA) to visually cluster present-day populations in Eurasia based on their allelic diversity, (Figure 1A). The diversity across PC1 indicates a cline in genetic variation stretching from West Eurasia (lower PC1 values) to East Asia (higher PC1 values).

The four newly reported Kuygenzhar individuals were projected onto the PCA, broadly forming two clusters (Figure 1A): B2_B5 and B2_B3, with lower PC1 values than the cluster conformed by B2_B6 and B2_B7. This distinct clustering of the samples suggests a possible difference in ancestry composition. B2_B6 and B2_B7 (who are father and son) plot closer to Mongolian and other eastern Asian genetic profiles, while B2_B3 and B2_B5 plot closer to other ancient Kazakhs and to present-day Kazakhs and Kyrgyz, suggesting they have a similar ancestry to these populations. There is a noticeable shift in ancestry of ancient medieval central Asian populations, which describe a cline from present-day Central Asian to Eastern Asian positions. The Kuygenzhar individuals are placed at the end of this cline described by ancient central Asian individuals.

Ancestry components were further analysed with the maximum likelihood clustering tool ADMIXTURE [50]. Following testing of a range of possible cluster (K) specifications, a K of 7 or 8 had the lowest mean cross validation error, indicating the best K for these populations. However, K = 5 provided a simpler model for data interpretation and visualisation while still achieving a low cross-validation error (Appendix A). The heterogeneity of the ADMIXTURE (K = 5) profiles for the ancient Kuygenzhar individuals (containing genetic affinities to all five ancestry components) suggest they are highly admixed (Appendix A).

#### 3.1.4. f-Statistics Ancestry Modelling

To reconstruct the demographic models giving rise to the ancestral composition of the newly reported Kuygenzhar samples, *qpAdm* tests were performed specifying both one-way and two-way models using a suite of published Central-Asian ancient individuals [7,8,11,57,58,59] (Appendix A); from similar geographic regions and time periods as the Kuygenzhar individuals in this study. No one-way models statistically fitted the ancestry of these individuals. However, one two-way model delivered good support both for each sample individually and for the samples grouped as single population (*n* = 4) (*p*-values > 0.05, Appendix A). This two-way model had Kyrgyzstan_Medieval_Nomad.SG (*n* = 3) and Kazakhstan_His.SG (*n* = 2) [11] as source populations, with the outgroups specified in Materials and Methods section [5,59]. Inferred ancestry proportions varied among the samples but the coefficient for Kazakhstan_His.SG was slightly higher in all of the models for the newly reported ancient samples and in the model with the four Kuygenzhar samples grouped as a single population (Appendix A). Kazakhstan_His.SG (which date from 1650–1850 CE) has a higher genetic affinity for West/Central Eurasia populations, while Kyrgyzstan_Medieval_Nomad.SG (1030–1278 cal CE) have higher affinities to East Asian populations.

### 3.2. Microbiome Analysis

#### 3.2.1. Red Complex Bacteria Detection and Authentication

Microbial metagenomic assignment of the DNA reads that were obtained from these individuals using Kraken2 identified many DNA reads that were assigned to bacteria frequently termed as members of the “red complex”: *Tannerella forsythia*, *Treponema denticola*, and *Porphyromonas gingivalis*; but also to other bacteria that are associated to periodontitis and oral cavity disease (Appendix A). Of these red complex bacteria, *T. forsythia* was the pathogen with a highest coverage (up to 11.4x in one individual) and was subsequently chosen for a more in-depth analysis (Table 2). For all Kuygenzhar samples, the non-human DNA was mapped against *T. forsythia* (92A2), *T. denticola* (ATCC35405), and *P. gingivalis* (ATCC33277) reference genomes. The end of the DNA reads showed the characteristic post-mortem damage in figures similar to those that were found in the host’s DNA, verifying the authenticity of aDNA (Figure 2A,B and Appendix A). The mapped samples differed in the coverage throughout the genome, being low coverage in B2_B5 and B2_B6 and high in B2_B3 and B2_B7 samples (approximately 11x and 9x, respectively) (Figure 3).

B2_B1 yielded a negligible amount of *T. forsythia* sequences. B2_B3 also yielded sequences at high coverage (relative to aDNA) for *T. denticola* and *P. gingivalis*; from B2_B7 we could also recover partial genomes of those bacteria (Appendix A).

For the B2_B3 sample, the proportion of positions of the reference genome that were covered (approx. two-thirds) was lower than expected for its high mean depth of coverage (Table 2, Figure 2C). This pattern is observed in other ancient and modern samples regardless of their geographical origin (Appendix A). Also, for this sample, the edit distances between the strain and the reference genome were larger than for the other three samples that were found (Figure 2D–G); the coverage was not evenly distributed across the genome, but in specific genes. This indicates that *T. forsythia* from B2_B3 might be a diverging strain, differentiated from that of the modern *T. forsythia* strains, or even a different, undescribed *Tannerella* species.

Aside from these differences, the similarities between the 18th century Kuygenzhar samples compared to the reference genome can also lead to interesting insights into the evolution of *T. forsythia*. For example, we detected an absence of sequencing coverage between positions ~1,315,000 and ~1,363,000 that is present in all samples that cannot be explained by sequencing fluctuations and may be the result of more recent evolution of this bacterium (Figure 3). This region, containing the genes *traJ*, *traK*, *traM*, *traN*, *rrf*, and *TetQ*, has previously been identified in ancient German and Mexican strains [15,74] and in contemporary strains it is present in highly varying proportions, ranging from 8% to 80% in Europe and North-America, respectively [81,82], while present in high frequencies in Latin America [15].

#### 3.2.2. Tannerella Forsythia Phylogeny and Clade Dating

A maximum likelihood (ML) phylogeny was performed on the 129,641 SNPs high-quality dataset with available ancient and modern strains to study the worldwide diversity of *T. forsythia*. Besides the Kuygenzhar strains, the analysis included two ancient German (950–1200 CE) strains, two ancient British strains (1770–1855 CE), two Pre-Hispanic Mexican strains (900 BCE–1200 CE), an ancient isolate from South Africa (1800), and thirteen modern strains, accounting for a total of 23 strains with an overall high bootstrap support (Figure 4 and Appendix A). The outgroup that was used to root resulting phylogenetic trees was a modern *Tannerella* strain that was isolated from a dog (OH2617_COT-023) and covered approximately 55% of the reference genome. Broadly, this pattern was previously described in a phylogenetic analysis of Mexican strains [15] although some samples cluster differently, possibly due to lower coverage and/or more-relaxed filtering noting that most nodes of the phylogeny have >90% bootstrap support for the nodes supporting ancient isolates (Appendix A).

To reconstruct the past and present diversity of *T. forsythia*, we built a second ML phylogenetic tree over the alignment including the lower coverage samples. The phylogeny placed the 18^th^ century Kazakhstan samples in two discrete phylogenetic clades with B2_B3 falling within ancient samples from Pre-Hispanic Mexico, and B2_B7 falling basal to the larger clade that includes most modern and ancient sampled diversity. When adding the low-quality strains to the tree we observe that there are three clearly distinct clusters (Appendix A); one, with high bootstrap support, encompassing most of the pre-Columbian and Kazakhs samples, but also one modern Spanish sample and one historical from Germany. A second cluster is composed of several Pre-Hispanic Mexico strains at the base (low bootstrap support) and a cline from base to tip of ancient to modern isolates. The third cluster includes the two historical high-quality English strains, the reference, and a number of isolates from diverse temporal and geographical origin very low support (Appendix A). This last cluster could be attributed to a low-coverage bias towards the reference.

In order to purge the phylogeny of events that could alter its topology and branch length, the alignment was screened for the presence of recombinant tracks. We identified 6003 recombinant tracks in the process (2,410,577bp; approximately 60% of the genome (Appendix A); see Methods) (Appendix A). In addition, we also removed a further 758 homoplasic sites, leaving 1,494 SNPs for further analysis.

After reconstructing the phylogeny using the filtered alignment, we obtained a phylogenetic tree with most of the distal branches well supported (Appendix A). Afterwards, we analysed the tree with *BactDating*, and we found evidence of statistically significant temporal signal in the phylogeny (R^2^ = 0.35, *p* value = 0.0071) (Figure 5A) (Appendix A). We estimated a mutation rate of 1.17 mutations per genome per year, and a most recent common ancestor (MRCA) for the clade around 8000 BCE (Figure 5B).

#### 3.2.3. Presence of Virulence or Lifestyle-Related Tannerella Genes

Kuygenzhar’s *Tannerella* strains were screened for possible virulence, antibiotic resistance, and glycosylation genes that were obtained from the literature [76] (Appendix A), along with ancient European and Mexican strains and the modern strains that were previously included in the phylogeny. Genes that were phenotypically associated to these functions may have endured differential selection pressures due to changes of environment in time and location, for instance the use of antibiotics and the change to a more sugar-rich diet [73]. All of the strains were mapped against the gene sequences and the resulting gene-by-gene coverage is shown in Appendix A. We note that a low overall mean coverage also results in a lower coverage of specific genes as evident in the Kuygenzhar samples B2_B5 and B2_B6. Also, the modern strains ASM386013v1 and ASM38536v1 have low coverage in all genes because they are divergent *Tannerella* like strains previously used as outgroups that share only approximately 55% of the reference genome (ASM23821v1) (NC_016610.1).

Regarding putative virulence genes, the Kuygenzhar samples show a lack of coverage over the protease *prtH* gene in three of the four Kuygenzhar samples with the exception of B2_B7. These proteases protect the bacterium by breaking down the bactericidal peptides and may contribute to pathogenicity by shedding active tumour necrosis factor from macrophage surfaces [83,84]. An association has been identified between the presence of *prtH* in *T. forsythia* and whether or not the bacterium causes periodontitis [85,86]. High levels of *T. forsythia* have been correlated with the presence of the *prtH* gene. *T. forsythia* infecting healthy subjects rarely possess the *prtH* gene, which suggest that this gene may be an indicator of periodontal disease. A low coverage was also detected in two other virulence genes, *tfsA* and *tfsB*; these genes are known to suppress the proinflammatory cytokine expression [87].

The antibiotic resistance gene *tetQ* (BFO-1235) is absent in all strains except the reference genome. This gene, located at approximately 1300Kb in the reference genome, is involved in the resistance against the tetracycline antibiotic. Its absence was previously observed in the historical German and Mexican samples [15,74]. Due to its absence in other modern strains from our dataset, it is possible that this gene has arisen under recent selective pressures [15,74]. In the Kuygenzhar strains there is a gap in the location of *tetQ* (BFO-1235) gene, further suggesting this genetic region may be a recent evolutionary addition.

There are two glycosylation genes, phosphatidyl-myo-inositol mannosyltransferase (*BFO-RS14425*) and *TIGR04157* family glycosyltransferase (*BFO-RS00485*), that are not detected in some Kuygenzhar strains while present in modern *T. forsythia*. These are both glycosyltransferases (GTFs) that are involved in the biosynthesis of O-glycan structures, occurring after the synthesis of glycan [76]. These genes are also absent in some ancient European strains, all ancient Mexican strains, as well as some modern strains. Again, this pattern may suggest a recent acquisition of glycosylation genes.

#### 3.2.4. T. denticola and P. gingivalis Trees

The phylogenetic trees that were generated for *T. denticola* (Appendix A) and *P. gingivalis* (Appendix A) also place the Kuygenzhar strains in basal branches of the modern strains for these bacteria, although in both cases additional ancient strains would be needed to clarify the evolution of these oral pathogens.

## 4. Discussion

Population genomics from human archaeological finds can help to obtain a better understanding about past migrations and population affinities [4,5,8]. By analysing both the human and non-human fraction of ancient DNA, we can explore these past people and the microorganisms that were associated with them [15,22,73,88]. In this research we have combined a human population genomics analysis with that of the oral bacterium *T. forsythia*.

The four Kuygenzhar individuals for which we obtained sufficient genetic data for downstream analysis lived during the Kazakh Khanate. Despite showing signals of two different ancestries, they bear general genetic similarities with present-day Kazakhs, as can be inferred by their close PCA position with present-day Kazakh and other central Asian populations (Figure 2A). As shown by qpAdm ancestry modelling, the newly reported Kuygenzhar samples’ ancestry could be fitted with a two-way model having two different historical central-Asian populations as sources. These source populations, although having a similar geographic origin, have fairly distinct ancestry; one carrying a West-Eurasian signal while the second yielding East Asian-like ancestry. This is clearly identified by their PCA positions [11] (Figure 1A). The literature suggests that East Asian ancestry-carrying groups of people moved into central Asia from present-day Mongolia after the collapse of the Golden Horde in the 14th century, which contributed to the present-day central Asian population’s ancestry composition [5,11,89]. This can also be observed by the cline that is described by ancient samples in the Eurasian PCA plot (Figure 1A) where the placement of ancient and historic diversity moves closer to eastern Asian positions through time. The position of B2_B6 and B2_B7 in this PCA are also displaced towards modern Eastern Asian ancestry as compared with the rest of the Kuygenzhar samples (Figure 1A). This is in agreement with the fact that the ancestry deriving from the East Asian-carrying source is slightly higher for both samples (Figure 1B; Appendix A).

The admixture composition that was found in the Kuygenzhar individuals, which cannot be modelled having a single preceding population as a source, agrees with the age of the remains that date shortly after the formation of the Kazakh Khanate [5,11]. Since the Kazakh Khanate started as a union of tribes, it is not surprising that the ancestry of the Kuygenzhar individuals is diverse. Although having an admixed genetic background (from two different populations), the Kuygenzhar were buried according to the Islamic ritual, suggesting that the Islamic religion could have acted as a unifying tool for the different groups of people that inhabited the region.

Population movements in the region could be associated with the rise and spread of diseases [90]. The oral infection periodontitis, caused by “red complex” bacteria, is a widespread disease that affects millions every year [91]. Recently, different studies have taken a look into this disease from a paleogenomic perspective, allowing for a better understanding of the bacteria involved [15,73]. However, *T. forsythia* bacterium, that was reclassified some decades ago [92], remains the most poorly characterized causative agent of periodontitis.

Previous microbial genomic studies have advantageously used dental calculus, which is the calcified plaque, as a source of oral pathogens [22,74]. However, calculus can be difficult to find in the archaeological record [93]. Our study indicates that dentine may be also a potential source for oral microbiome DNA, which may increase the range of potential data for this type of research. Of the four Kuygenzhar individuals, two had high amounts of *T. forsythia* on their teeth, an observation that is consistent with periodontitis. Due to lower-than-expected mapping, as well as a patchy mapping against the reference genome, we suspect that B2_B3 contains a different and more divergent strain of *T. forsythia* compared to the reference genome. The other sample with high *T. forsythia* levels, B2_B7, had a high mean coverage and a large part of the reference covered.

Firstly, a phylogenetic analysis was performed to infer the evolutionary history of the Kuygenzhar strain in context of the worldwide diversity. This analysis proved complicated due to the absence of a clear outgroup. We have used a *Tannerella*-like strain as an outgroup, covering approximately 55% of the reference genome. The tree was reasonably supported; although due to a low number of SNPs remaining after homoplasy and recombination removal make it difficult to assign a proper mutation rate and confidence intervals for the different nodes. This may be due to a high diversity within the classified *T. forsythia* strains as well as a lack of a high-quality dataset. Nonetheless, it seems clear that there are two main branches in the *Tannerella* phylogeny, one containing mostly modern and historical samples that are fairly similar to the reference assembly, and another cluster that is made up entirely of ancient strains, including our Kazakh isolates. In particular, this clade seems to be fairly distant, genetically speaking, to present day human *Tannerella* strains, but could be similar to other non-yet sampled *Tannerella* strains.

Broadly, this analysis indicates similarities between Colonial Mexican and European strains which are more closely related to the modern strains. Also, Pre-Hispanic Mexican and Kuygenzhar strains may be more closely related. However, the age and the coverage of the ancient strains may have influenced this structure, which should be considered with caution, particularly considering the limited sampling of modern diversity. Future research with a high-quality dataset containing ancient strains from all around the world will be able to provide a clearer view of the diversity of *T. forsythia*.

The evolutionary history of *T. forsythia* was explored by analysing the coverage of putative virulence, antibiotic resistance, and glycosylation genes. The *prtH* gene, which has been associated with virulence, and the *BFO-RS14425* and *BFO-RS00485* glycosylation genes were not detected in some Kuygenzhar individuals, while the BFO-1235 (*tetQ*<) antibiotic resistance gene had no coverage in all 18^th^ century strains. This differential presence in the Kuygenzhar as well as the other ancient strains indicates which genes may have evolved more recently due to changes in environmental pressures. One example of possible recent evolution is BFO-1235 (*tetQ*), which is an antibiotic resistance gene which acts to protect the bacterial ribosome from binding to tetracycline, making it resistant to this antibiotic [15]. This gene is present in the modern reference genome and may have been the result of recent evolution since it is present in highly varying proportions across the world [15]. The virulence gene, *prtH*, was only detected in one of the Kuygenzhar strains, B2_B7. This gene has been correlated with the presence of periodontitis [85,86]. Together with the high levels of *T. forsythia*, this suggests that the individual B2_B7 may have suffered from periodontitis, giving insight into the health conditions of this individual. Interestingly, no morphological evidence of caries or dental abscesses in the dentition of this individual were observed.

Meat and dairy products were the basis of nutrition in the traditional Kazakh society of the 17th–19th centuries. The meat ration was mainly represented by horse and lamb, more rarely by beef and camel. In the mobile communities of cattle breeders of the late 19th–early 20th centuries, milk and dairy products were the basis of the population’s nutrition, accounting for 80–90% of the total balance of food that was consumed by humans [76]. Dairy food was especially varied, which is considered as evidence of its ancient origin. The centuries-old livestock breeding contributed to the gradual development by the Kazakhs of a wide range of dairy products, which were consumed mainly in spring and summer during the period of maximum milk yield. Various dairy products were also prepared and conserved for the winter when the milking of cattle stopped [77]. The Kazakh population of the 18th–19th centuries consumed milk from mares, camels, cows, sheep, and goats [78]. Fish consumption in the Sary-arka region of Kazakhstan (the geographical region around the present-day capital of Kazakhstan, Nur-sultan) was low tending to be clustered in geographical regions with large water bodies such as the areas around the Korgalzhin and Tenghiz lakes (South-west of Nur-sultan) as well as Siletyteniz, Kyzylkak, Zhalauly, and Teke lakes (North-east of Nur-sultan) [76]. The widespread consumption of bread by the Kazakhs, like other cultivated cereals, started after the colonization of the Russian Empire and the resettlement of Russian peasants from the second half of the 19th century. Until that time, bread and its food consumption in general were familiar only to the wealthy top of the nomadic society, while most of the steppe population did not use bread from cereals at all in everyday life [79].

In conclusion, 18th-century Kazakh Khanate individuals were still relying on a traditional steppe diet, strongly based on meat, milk, and dairy products from horses, sheep, and goats. Therefore, the partial lack of *prtH* gene and the absence of the two glycosylation genes can be due to a recent evolutionary change that was associated with a subsequent increase in sugar in the diet [73].

## 5. Conclusions

This study provides new insights into the genetic composition and ancestry of the 18th-century Kazakh Khanate people. Genomic analyses demonstrates that their ancestry likely reflected an East Asian admixture from the Mongol invasion that occurred in medieval times. Furthermore, we provide an understanding of the health status of the Kuygenzhar individuals by researching *T. forsythia* bacterial DNA that was recovered from their oral microbiome. Of the individuals, two had ah high presence of this bacterium and may have had periodontitis, shedding new light on their health status. Further, our study demonstrates that, even with a diet based on meat and dairy products -and likely very low carbohydrate intake- individuals could develop periodontal disease. Observations of ancient *T. forsythia* provide direct evidence on when modern diversity originated and dispersed. In addition, it can determine the temporal emergence of genes of phenotypic relevance which can also provide useful information for future clinical research. This kind of human-pathogen data needs to be integrated in a broader archaeological, historical, and anthropological context to understand past human migrations and their adaptation to different diets and environments.

## 6. Patents

No patents have been generated.

## Figures and Tables

**Figure 1 biology-10-01324-f001:**
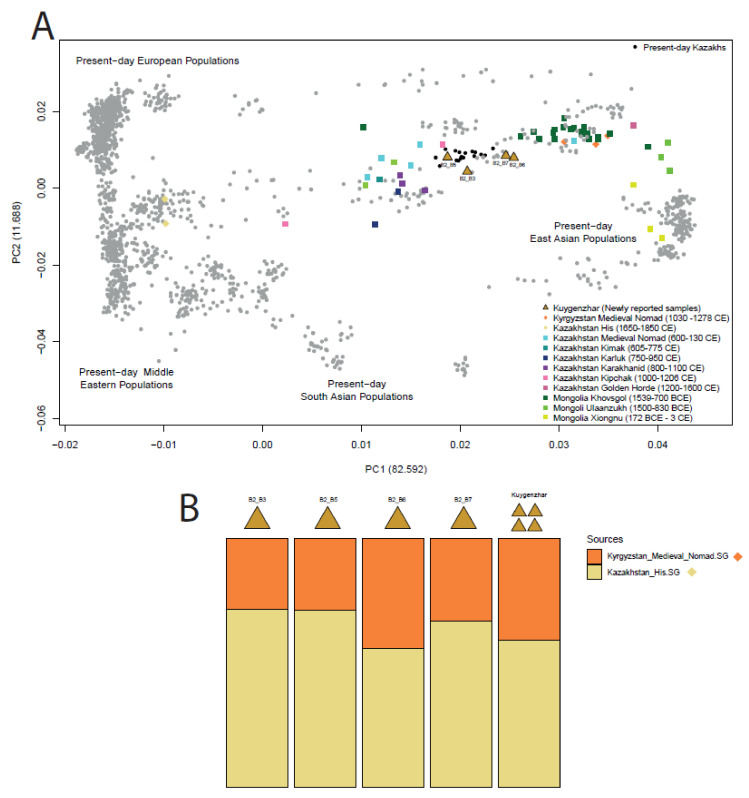
(**A**) Principal component analysis (PC.A) of the genetic variation in Eurasia (grey points) with ancient samples projected onto this. The Kuygenzhar individuals (golden triangles) are clustering near to present-day Kazakhs. (**B**) Ancestry modelling performed with qpAdm for the fitting two-way model with a *p*-value > 0.05 for all individuals individually and grouped.

**Figure 2 biology-10-01324-f002:**
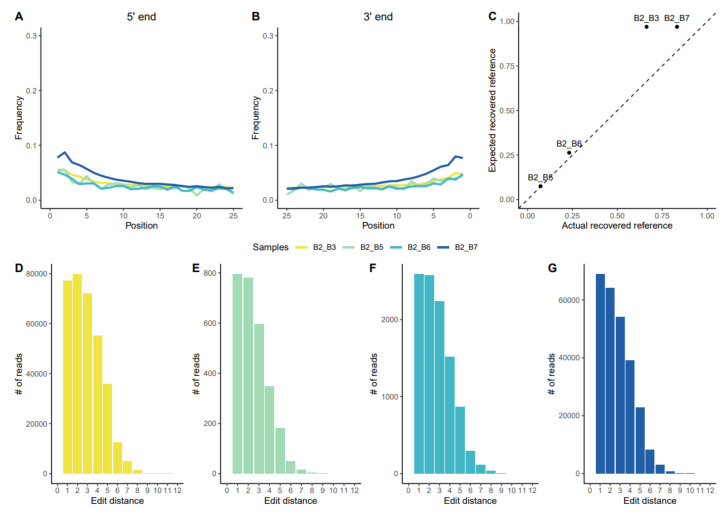
(**A**,**B**) Post-mortem damage patterns at the 5′ and 3′ends for the four samples with more than 0.05x coverage mapped against the *T. forsythia* reference genome (B2_B3 yellow, B2_B5 green, B2_B6 cyan, and B2_B7 blue). (**C**). Expected percentage of the *T. forsythia* reference genome that was recovered against the actual recovered percentage for the analysed samples. (**D**–**G**) Sequences’ edit distance of each different sample mapped against the *T. forsythia* reference genome (B2_B3, B2_B5, B2_B6, and B2_B7; in that order, following colour codes).

**Figure 3 biology-10-01324-f003:**
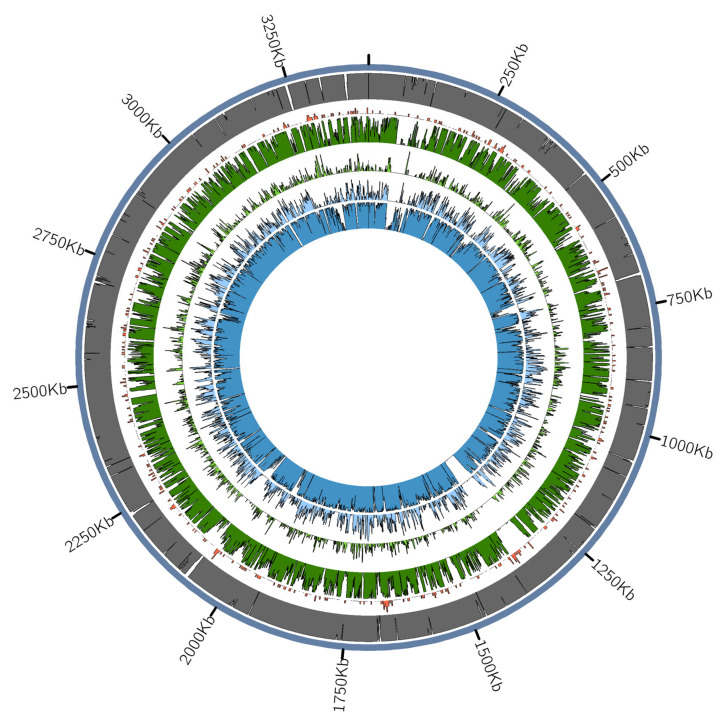
Mapping of the four 18th century Kuygenzhar samples against the *Tannerella forsythia* reference genome (NC_016610.1). The outer ring (**blue**) shows the reference genome mapped against itself, the presence of genes is indicated in orange, and the four Kuygenzhar samples from the inner rings in the order of: B2_B3 (**dark green**), B2_B5 (**light green**), B2_B6 (**light blue**), and B2_B7 (**dark blue**).

**Figure 4 biology-10-01324-f004:**
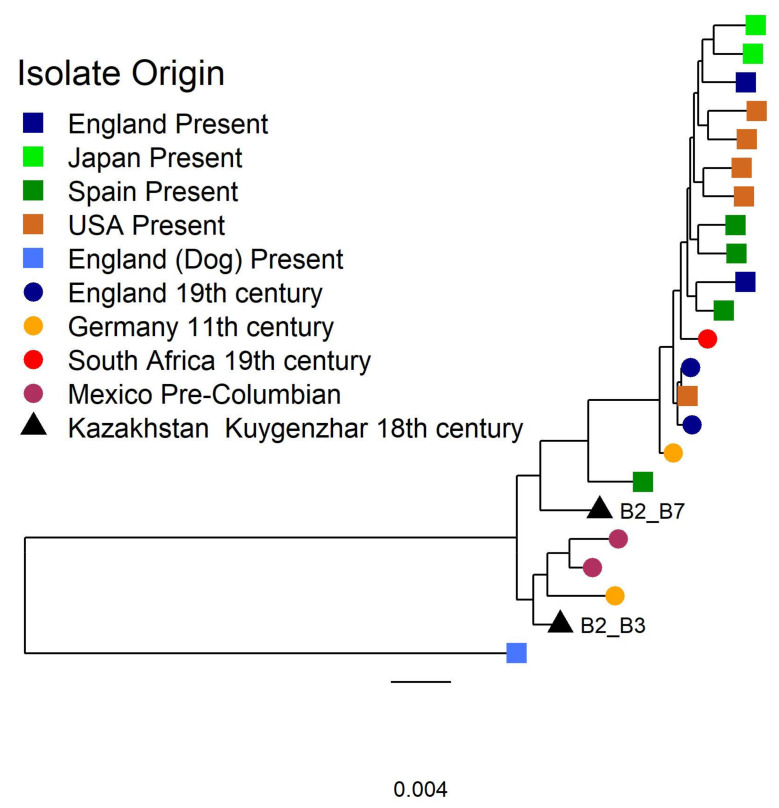
Phylogenetic tree of the high-quality alignment covering 129,641SNPs for *Tannerella forsythia*. Isolates present in the tree include 13 human modern isolates from Europe, America, and Japan; seven ancient samples from Africa, Europe, and America, our two isolates from Kazakhstan (triangles); and a modern isolate recovered from a dog used as a phylogenetic outgroup.

**Figure 5 biology-10-01324-f005:**
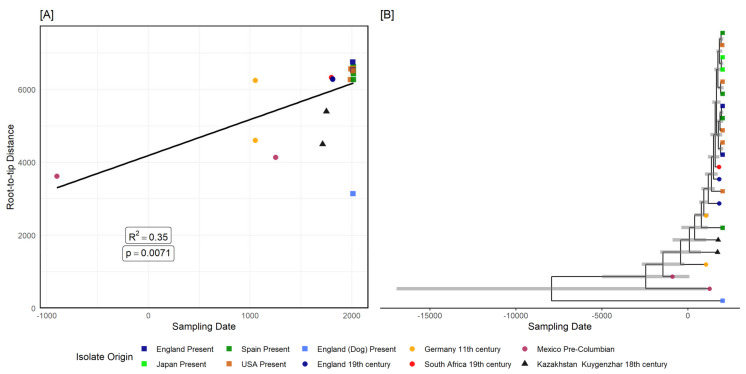
Tip calibrated phylogeny of *Tannerella forsythia*. Root–tip distance analysis of the analysed samples after recombination and homoplasy pruning, using the tip distance to the root (*y*-axis) and the collection time of the isolate (*x*-axis) (**A**). The ML phylogenetic tree created with 1494 recombination and homoplasy free positions, with temporal annotation data (higher posterior densities) for each node (**B**).

**Table 1 biology-10-01324-t001:** Mapping statistics for the Kuygenzhar samples that were mapped against the human reference genome GRCh37. B2_B1 was removed from further analyses due to insufficient data.

Nuclear DNA	Number of Sequenced Paired Reads	Mapped Reads	Reads with a Mapping Quality > 30	Average Coverage	Covered Positions (%)
B2_B1	51,212,164	6715	5140	0.0002x	0.02%
B2_B3	86,182,644	13,108,305	11,448,368	0.3977x	31.59%
B2_B5	82,504,751	16,069,879	13,838,347	0.4788x	36.18%
B2_B6	54,440,203	13,936,061	11,947,290	0.3379x	26.49%
B2_B7	40,649,429	3,347,196	2,961,184	0.0973x	9.12%
Mitochondrial DNA					
B1_B2	51,212,164	41	39	0.2187x	19.05%
B2_B3	86,182,644	8799	6847	43.8743x	99.99%
B2_B5	82,504,751	18,689	12,439	82.3307x	100%
B2_B6	54,440,203	7853	6212	33.8299x	99.98%
B2_B7	40,649,429	2215	1991	11.0503x	99.79%

**Table 2 biology-10-01324-t002:** Mapping statistics of Kuygenzhar non-human reads mapped against the *Tannerella forsythia* reference genome (NC_016610.1).

Sample Name	Human Free Reads	Mapped Reads	Reads with a Mapping Quality > 30	Average Coverage	Covered Positions (%)
B2_B1	148,214,698	273	132	0.003x	0.31%
B2_B3	87,420,490	537,695	391,726	11.416x	66.29%
B2_B5	84,107,411	4275	3366	0.081x	7.23%
B2_B6	114,507,988	15,450	12,304	0.328x	23.18%
B2_B7	117,513,937	428,178	318,145	8.889x	83.23%

## Data Availability

The data presented in this study are available at the European Nucleotide Archive (ENA): PRJEB49091.

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
