# Peer review of "Genomic Analysis of 18th-Century Kazakh Individuals and Their Oral Microbiome"

_biology, 2021, doi:10.3390/biology10121324_

Round 1

Reviewer 1 Report

The manuscript is very well written, methods and results are sound. It was a pleasure to read it because of its clarity. The combined analysis of human and microbiota genoms provide an interesting complete description of these samples.

I do not have particular changes to suggest, I only have a doubt about the choice of term haplotype/haplogroups in line 312 (“all observed haplotypes can be considered common in sampled current-day Kazakh populations”, would you mean haplogroups instead of haplotypes?) and Table S4 (it would be haplogroups instead of haplotypes as well?).

Line 322: would you mean haplogroups instead of haplotypes?

Author Response

The reviewer is right; in lines 312, 322 and Table S4, it should be "haplogroups" instead of "haplotypes"

Reviewer 2 Report

This article describes the genetic analyses of four individuals (B2-B3, B2-B5, B2-B6 and B2-B7) who lived in Kazakhstan during the Kazakh Khanate, in the early to mid-18th century. The authors determined that, as in present-day populations, the ancestry of the 4 individuals from Kuygenzhar was diverse. The manuscript also describes the DNA analyses of some oral bacteria preserved on the individuals’ dental calculus and linked to periodontitis. One individual (B2-B3) was found to have a divergent strain of Tannerella forsythia.

First, I want to emphasize that I am not a microbiome expert, so my review will focus on the ancient DNA study. Overall, the manuscript is very well written and interesting. I really appreciated the detailed description of the methods, especially the paragraphs on bioinformatics tools. It seems that these days, ancient DNA scientists need to have more knowledge in bioinformatics than in molecular biology. The number of programs that was used in this study is truly impressive.

Here are my comments

  • It would be nice to have a bit more information about the ages of the individuals when they died. When you say that B2-B6 is the father of B2-B7, do the anatomic analyses of the skeletons agree with this assertion? I would have expected the older individual to have more bacteria than a younger individual. Yet the coverage for T. forsythia is lower for B2-B6 than it is for B2-B7. Could B2-B6 be the son instead of the father? I agree that the state of preservation could explain this difference, so it may be interesting to look at the proportions of human to bacterial DNA for these two samples.
  • Here are 3 errors that should be addressed: Sup Table S7 and S8 are missing; on line 408, Figure 1: D, E, F, G should be changed to Figure 2: D, E, F, G; Table 2 is presented before Table 1. It should be moved to the Results section.
  • My two structural recommendations would be to 1- clearly separate the analyses done on human DNA from those performed on bacterial DNA in the Results section, and 2- to avoid repeating the processing of sequences in 2.2 and 2.9, unless there are differences (different seedling maybe?).
  • Figure 2 is unclear. Since the MapDamage profile of the human DNA is shown on Figure S3, those from P. gingivalis and T. denticola are shown on Figure S7 and 8, I am guessing that Figure 2A, B show the damage detected in T. forsythia. It should be clarified in the legend. Same comment for Figure 2C. Finally, I do not understand what Figure 2: D, E, F, G represents. Could you clarify it a bit in the text?
  • Please clarify: was the same 50 mg of cementum used for both human and bacterial DNA analyses? Was there only one DNA extraction per individual?

Minor cosmetic comments

  • In Table 2 (and in the text), the depth coverage is given with a number followed by an upper-case X. I would change that to a lower-case x.
  • Figure 6 is a bit fuzzy. It is also bulky. Maybe it should be moved to the Supp material?
  • This sentence should be improved: “The original kurgan was disturbed atop to accommodate the graves of seven people between 0.85 and 1.15 metres depth adhering to the Islamic ritual”.
  • Line 398: Tannerella forsythia should be in italics
  • Line 642: These kinds of human pathogen data need to.. instead of needs to. Or This kind of human pathogen data needs to.
  • The sentence that starts on line 639: In addition, ancient observations… is clumsy. It should be rephrased.
  • Line 649: couldn’t find “Tables S.7 to S.11 (one document for each).”
  • What do numbers 1-5 on Figure S1A represents? Do they show where the skeletons were found?

Bibliography

  • The bibliography is incorrectly formatted (the publication year should be listed in bold, after the journal’s name).
  • Line 289: there is no reference for iTOL.
  • Reference 43 seems incomplete. Maybe add the GitHub link?
